# Electromagnetic characteristics of ENSO

Johannes Petereit [1,2], Jan Saynisch[1], Christopher Irrgang[1], Tobias Weber[1], and Maik Thomas[1,2]

[1]GFZ German Research Centre for Geosciences, Potsdam, Germany
[2]Freie Universität Berlin, Institute of Metereology, Berlin, Germany

**Correspondence:** Johannes Petereit (petereit@gfz-potsdam.de)

**Abstract.** The motion of electrically conducting sea water through Earth's magnetic field induces secondary electromagnetic fields. Due to its periodicity, the oceanic tidally-induced magnetic field is easily distinguishable in magnetic field measurements and therefore detectable. These tidally-induced signatures in the electromagnetic fields are also sensitive to changes in oceanic temperature and salinity distributions. We investigate the impact of oceanic heat and salinity changes related to the El Niño/Southern Oscillation (ENSO) on oceanic tidally-induced magnetic fields. Synthetic hydrographic data containing characteristic ENSO dynamics have been derived from a coupled ocean-atmosphere simulation covering a period of 50 years. The corresponding tidally-induced magnetic signals have been calculated with the 3D induction solver x3dg. By means of the Oceanic Niño Index (ONI), based on sea surface temperature anomalies, and a corresponding Magnetic Niño Index (MaNI), based on anomalies in the oceanic tidally-induced magnetic field at sea level, we demonstrate that evidence of developing ENSO events can be found in the oceanic magnetic fields statistically 4 months earlier than in sea surface temperatures. The analysis of the spatio-temporal progression of the oceanic magnetic field anomalies offers a deeper understanding on the underlying oceanic processes and is used to test and validate the initial findings.

## 1 Introduction

The El Niño/Southern Oscillation (ENSO) is well known for its warm and cold temperature anomalies caused by changes in the ocean-atmosphere system in the equatorial Pacific Ocean. These anomalous events, known as El Niño and La Niña, cause extreme weather conditions throughout the globe, e.g., tropical cyclones (Vincent et al., 2011), droughts, bush fires and floods (Philander, 1983). The extreme weather affects entire ecosystems (Glynn and De Weerdt, 1991) and causes damages to infrastructure and agricultural production (Wilhite et al., 1987). Substantial socio-economic costs are the consequence. The prospective doubling of extreme El Niño events, as a response to greenhouse warming (Cai et al., 2014), would increase the socio-economic costs even further. The negative impacts can be mitigated with pre-emptive measures provided reliable El Niño forecasts are available.

Knowledge about spatio-temporal variations of upper-ocean heat content, aquired traditionally by moorings, are a major source of ENSO predictability (Meinen and McPhaden, 2000). Monitoring of seawater temperature and salinity anomalies are consequently a prerequisite for improved ENSO forecasting, especially since changes in thermocline depth, caused by equatorial Kelvin waves, have been known to precede sea surface temperature anomalies (Harrison and Schopf, 1984). These anomalies are already measurable with an array of moored buoys (TAO/TRITON) which monitors the upper ocean temperature.

Furthermore, measurement of these thermocline displacements with altimetric methods has been subject of extensive research (Ji et al., 2000; Picaut et al., 2002, 1996).

Changes in the oceanic heat content can also be infered from the motion-induced electromagnetic fields of the ocean (Minami, 2017; Saynisch et al., 2016). This new and lesser known method complements the pre-existing techniques. It can be applied to detect mainly large-scale oceanic processes altering temperature and salinity distributions.

The flow of electrically conducting seawater generates an electric current due to the interaction of moving salt ions with the geomagnetic field. These electric currents induce a magnetic field with a local magnitude of several nano Tesla (nT) (Maus and Kuvshinov, 2004).

The magnitudes of ocean flow and electric seawater conductivity determine the oceanic magnetic field strength in dependence of the field strength of the ambient geomagnetic field. The ocean flow is classically divided into general ocean circulation and ocean tides.

The ocean circulation driven by momentum and buoyancy fluxes is irregular in time. Its effect are consequently difficult to separate from magnetic field measurements. However, the circulation-induced magnetic field's non-trivial contributions to the geomagnetic field have been subject of many studies (Tyler and Mysak, 1995; Manoj et al., 2006; Irrgang et al., 2016a, b). Manoj et al. (2006) analysed the influence of changes in the equatorial current system caused by ENSO, onto the circulation-induced magnetic field. They neglected temporal variations in oceanic temperature and salinity and assumed a time-constant oceanic conductivity distribution. In their study, the difference between the global circulation-induced magnetic field during normal conditions and El Niño conditions was estimated. They found that ENSO related magnetic field anomalies in the equatorial region were too small to be distinguishable from the magnetic field anomalies caused by variations of the Antarctic circumpolar current (ACC). The large ACC anomalies extend into the equatorial region and over-shadow the small effect of ENSO. The magnitude of the ACC anomalies was found to be of $\pm 0.2$ nT at Swarm altitude (430 km) in the equatorial Pacific and can therefore be assumed to be an upper limit to ENSO's circulation-induced magnetic field anomalies.

The periodicity of the tidal flow on the other hand allows an easy separation of its magnetic field from other constituents in geomagnetic field measurements. Its signals have been extracted successfully for the semidiurnal M2 and N2 tides from measurements of the magnetic satellite missions CHAMP and Swarm (Tyler et al., 2003; Sabaka et al., 2016). Amplitude variations of these periodic magnetic signals are mainly caused by variations in seawater conductivity distribution (Saynisch et al., 2016). Seawater conductivity is sensitive to seawater temperature and salinity. In comparison to the amphidromic system of the tides, these quantities exhibit high temporal variability. Consequently, the majority of information gained from anomalies of the oceanic tidally-induced magnetic signals are linked to changes in oceanic temperature and salinity distributions. Modeled and measured tidally-induced magnetic fields are in good agreement (Tyler et al., 2003; Maus and Kuvshinov, 2004; Sabaka et al., 2016), offering the possibility for *in silico* sensitivity studies.

The influences of global climate variations, such as Greenland glacial melting and global warming, on the electromagnetic oceanic tidally-induced signals have already been investigated (Saynisch et al., 2016, 2017). For these cases, the tidally-induced radial magnetic field was found to be an appropriate measure to monitor climate variations of the global oceanic conductivity on decadal time scales.

With a period of 4-7 years, ENSO is acting on monthly to annual time scales. And despite its impact on the global climate, the immediate climatological influences on the ocean are limited to the region of the equatorial Pacific. ENSO's characteristic changes in the ocean circulation alter the Pacific upper ocean temperature and salinity distributions ($\approx 300$m) within months.

In our study, we follow the approach of Saynisch et al. (2017) and investigate whether the electromagnetic oceanic tidally-induced signals could be used as an appropriate measure for ENSO induced changes in seawater conductivity and, consequently, the different stages in the dynamics of the El Niño/Southern Oscillation.

In section 2, we introduce the oceanic electromagnetic induction along with the models and data used to compute the resulting climate sensitive electromagnetic oceanic tidally-induced signals (EMOTS). Also, based on the radial magnetic component of the modelled EMOTS, we define a Magnetic Niño Index. In section 3, we compare the classical Oceanic Niño Index (ONI) to the proposed Magnetic Niño Index and discuss differences and similarities. We conclude and summarize the study in section 4.

## 2 Models and Data

### 2.1 Ocean data

We simulated ENSO with the ECHAM6/MPIOM a global coupled atmosphere-ocean general circulation model (AOGCM) (Giorgetta et al., 2013).

The Max-Planck-Institute Ocean Model (MPIOM, (Jungclaus et al., 2013)) is a general ocean circulation model. The model solves the primitive equations for a hydrostatic Boussinesq fluid on a curvilinear Arakawa-C-grid with poles shifted to Antarctica and Greenland. The ocean is discretised on a grid with a horizontal resolution of $\sim 3.0° \times 1.8°$ (GR30) and an irregular vertical distribution over 40 horizontal levels.

The atmosphere general circulation model ECHAM6 (Stevens et al., 2013) is applied with the horizontal resolution of $\sim 3.75° \times 3.75°$ (T31) and 31 vertical hybrid sigma/pressure levels.

The simulated ocean data covers 50 years of monthly mean seawater temperature $T$, seawater salinity $S$ and seawater pressure $P$. Using the Gibbs seawater equation (TEOS-10 (IOC et al., 2010)), the electric seawater conductivity $\sigma$ can be calculated from, $T$, $S$ and $P$. Present-day conditions were used to run the coupled AOGCM in a free mode instead of a observation-driven forcing. Therefore, the modeled climate represents reality only in a statistical sense.

### 2.2 Oceanic tidally-induced currents and EMOTS

The tidally-induced electric current, the source for the electromagnetic oceanic tidal signals (EMOTS), is derived for each month with the following two step algorithm.

First, the product of seawater conductivity $\sigma$ and tidal velocities $\boldsymbol{v}_{\mathrm{M2}}$ is integrated from ocean bottom (-H) to surface (SSH)

$$\boldsymbol{V}_{\mathrm{M2}}(\varphi, \vartheta, t) = \int_{-H}^{SSH} \sigma(\varphi, \vartheta, z, t) \cdot \boldsymbol{v}_{\mathrm{M2}}(\varphi, \vartheta, z, t)\, dz, \tag{1}$$

where $\varphi$, $\vartheta$ and $z$ are longitude, co-latitude and depth. The tidally-induced electric current $\boldsymbol{j}_{M2}$ is then calculated as the cross-product of the depth-integrated and conductivity-weighted transports $\boldsymbol{V}_{M2}$ and the ambient geomagnetic field $\boldsymbol{B}_{Earth}$ as

$$\boldsymbol{j}_{M2}(\varphi,\vartheta) = \boldsymbol{V}_{M2}(\varphi,\vartheta) \times \boldsymbol{B}_{Earth}(\varphi,\vartheta). \tag{2}$$

Variations in the amphidromic system are negligible even on decadal time scales (Saynisch et al., 2016). Consequently, we
followed the approach of Saynisch et al. (2017) and assumed the tidal system to be invariable in time. Tidal amplitudes and phases of the oceanic M2 tide were taken from the TPXO8-atlas (Egbert et al., 1994; Egbert and Erofeeva, 2002).

For this study, the geomagnetic field $\boldsymbol{B}_{Earth}$ was estimated with the International Geomagnetic Reference Field edition IGRF-12 (Thébault et al., 2015). Our study focuses on the effects of oceanic conductivity variations. $\boldsymbol{B}_{Earth}$ is consequently assumed to be constant in time. Naturally occurring secular variations in $\boldsymbol{B}_{Earth}$ will linearly vary with $\boldsymbol{j}_{M2}$ (eq. (2)). Since the variations
in the geomagnetic field are well known for near real-time observations (Gillet et al., 2010), its effects can be removed before analyzing observational data of EMOTS for the influence of ENSO.

The electric current $\boldsymbol{j}_{M2}$ oscillates with the rise and fall of the tidal velocities. The time-variable magnetic field associated with $\boldsymbol{j}_{M2}$ interacts with the electrical conducting environment. Due to secondary effects, additional electrical currents and electromagnetic fields are induced. The solution to the posed induction problem, the entirety of the resulting electromagnetic
fields, are called electromagnetic oceanic tidal signals (EMOTS) (Saynisch et al., 2017). In our study, they are computed at sea level with the 3D EM induction solver x3dg (Kuvshinov, 2008). The solver is based on a contracting volume integral equation approach (Pankratov et al., 1995; Singer and Fainberg, 1995).

To realistically model EMOTS, Earth's electrical mantle conductivity and the electrical oceanic conductivity need to be included in the model-setup (Grayver et al., 2016). The mantle conductivity is represented by a time-constant 1-D spheri-
cal symmetric conductivity distribution following Püthe et al. (2015). The time-variant ocean conductivity and the constant sediment conductance, i.e., depth-integrated conductivity, are represented by an inhomogeneous spherical conductance layer situated on top of the mantle conductivity. This conductance layer combines sediment conductance and modeled monthly mean ocean conductance, derived from modeled $T$, $S$ and $P$. The sediment conductance is a combined result of the method of Everett et al. (2003) with the global sediment thickness of Laske and Masters (1997).
The strict periodicity of the tidal induction process, and therefore the resulting EMOTS, allows an easy extraction of these signals from real observations. Geomagnetic observations naturally contain contributions of several magnetic signals and are consequently noisy. Given sufficiently long time series of measurements, fitting of a specific frequency allows to separate even weak signals. Consequently, other non-periodic oceanic processes such as equatorial Kelvin waves or tropical instability waves are not directly detectable in noisy observations although their flow is also creating an magnetic field. But, the changes in
temperature and salinity distributions caused by variations in thermocline depths or travelling patterns of cold and warm water fronts are detectable through variations in the EMOTS.

In general, EMOTS allow to infer information of oceanic temperature and salinity distributions throughout the whole water column due to the integrative nature of the induction process. On annual and decadal time scales, the variabilty in the geomag-

netic field and the amphidromic system are small compared to the variability of oceanic temperature and salinity distributions. Consequently, the variability in $j_{M2}$, and therefore the resulting EMOTS, are mainly caused by changes in the electrical conductivity. The processes in question, however, need to take place on time scales longer than the tidal waves in order to be detectable.

Comparable findings are to be expected for all magnetic and electric field components that form the entirety of EMOTS. However, in this study we focus solely on $B_r$, the radial component of the magnetic field of the EMOTS, at sea level. Out of the magnetic components it is the only one measurable outside of the ocean (Chave and Luther, 1990). $B_r$ has been observed succesfully with magnetic satellite missions such as CHAMP (Tyler et al., 2003) and Swarm (Sabaka et al., 2016). Compared to the signal strength at satellite altitude, the signal is approximately 30% stronger at sea level. Additionally, substantial research

has already been conducted to investigate the relation between oceanic induced magnetic field variations, especially their radial component, and their oceanic causes (Saynisch et al., 2016, 2017; Irrgang et al., 2016a, 2017, 2018; Saynisch et al., 2018). We add to this canon by investigating the impact of ENSO, the biggest interannual climate signal, onto the oceanic tidally-induced radial magnetic field.

## 2.3    Indices and statistical analysis

Different indices have been used to characterize ENSO events (Hanley et al., 2003). A current state-of-the-art indicator is the Oceanic Niño Index (ONI) (NOAA, 2017) from the climate prediction center of the National Oceanic and Atmospheric Administration (NOAA). The ONI is used to monitor the oceanic part of the ocean-atmosphere phenomenon. It is defined as a 3 months running mean of sea surface temperature anomalies in the Niño 3.4 region (i.e., 5°N - 5°S, 120°W - 170°W) relative to the mean annual signal of regularly updated 30-year base periods. Warm and cold events are identified as periods exceeding

a threshold of $\pm0.5$°C longer than 4 months. The sea surface temperatures of the Niño 3.4 region have been known to correlate well with ENSO (Bamston et al., 1997).

     In our study, the ONI calculations are based on the data of the model climate exeriment conducted with the ECHAM6/MPIOM. Since no significant trends are present in our data, we used all 50 years as a base period for the ONI calculation, instead of the running 30 year base period used by NOAA.

We also calculate a comparable index based on the radial tidally-induced magnetic field $B_r$ (see section 2.2), the Magnetic Niño Index (MaNI). The same algorithm as in the ONI calculation is used with the difference that the sea surface temperature anomalies are substituted with $B_r$ anomalies in the Niño 3.4 region.

     The relation of the indices is analysed by calculating their correlation. A time delay analysis is carried out by calculating and analysing the cross-correlation. For two time series, the cross-correlation is the evolution of correlation between those two

when they are shifted against each other in time. It is used to identify temporally lagging or leading signals.

## 3 Results and Discussion

### 3.1 Comparison of derived ENSO indices

From our modeled data, we derived two indices (see sec. 2.3). First, following the algorithm of the National Oceanic and Atmosphere Administration (NOAA), we calculated the classical Oceanic Niño Index (ONI) from seasurface temperatures (SST). Then, we adapted the algorithm for the modeled tidally-induced magnetic fields and created a Magnetic Niño Index (MaNI). The time series of both indices are shown in figure 1. In agreement with the NOAA classification (NOAA, 2017), 7 El Niños and 10 La Niñas can be found in the climate model data. Following Null (2017), 1 out of the 7 El Niños is classified as very strong ($\geq 2.0°$ C), 3 are found to be moderate (1.0 to $1.4°$ C) and 3 are classified as weak (0.5 to $0.9°$ C). The set of La Niña events consists of 6 moderate ($-1.0$ to $-1.4°$ C) and 4 weak events ($-0.5$ to $-0.9°$ C).

The strongest El Niño, the very strong warm event, is found at the most prominent peak of the time series. Starting at month 133 of the modeled time period, it lasts 16 months and reaches a maximum value of $2.3°$ C (figure 1). These values are comparable to that of the El Niño events taken place in winter 1997/8 or 2015/6, with anomalies of $2.3°$ C and durations of 13 and 19 months, respectively (NOAA, 2017).

The spatially averaged temporal mean radial oceanic magnetic field amplitude ($B_r$) in the Niño 3.4 region was found to be 0.546 nT. The mean seasonal variation obtained by the climatology is $\pm 0.29$ pT (pikoTesla) and is three orders of magnitude smaller than the mean signal. The Magnetic Niño Index (MANI) based on $B_r$ anomalies relative to the 50 year climatology at sea surface height, has a range of $-0.84$ pT to 0.82 pT which is in the same order of magnitude as the seasonal variation.

While the ONI covers the development of sea surface processes, the MaNI also includes subsurface processes. $B_r$ is an integral measure incorporating the seawater conductivity integrated from ocean bottom to sea surface (see eq. (1) and eq. (2) in sec. 2). Despite their different perspectives on oceanic processes, both indices show a correlation of 0.63. The SST based index ONI is used to quantify the duration and strength of anomalous ENSO events. The high correlation of both indices shows that ENSO's effects have considerable impact on sea surface processes (ONI) as well as subsurface processes integrated in the tidal magnetic field.

The analysis of the cross-correlation of the two indices (embedded plot in figure 1) shows a MaNI-lead of 4 months over the ONI. Accounting for this lead, the correlation of both time series increases to 0.72.

Since in our setup (see sec. 2.2) the only time-variable contribution to $B_r$ is the seawater conductivity $\sigma$, we conclude that in the Niño 3.4 region subsurface anomalies of $\sigma$, caused by anomalies in $S$ and $T$, are leading SST anomalies.

Current magnetometers like the absolute scalar magnetometers of the Swarm mission with accuracy of $< 45$ pT and a sensitivity of $1$ pT$/\sqrt{\text{Hz}}$ (Jager et al., 2010) are able to resolve the global structure of the oceanic tidally-induced radial magnetic field. To be able to observe the presented variations an increase magnetometer precision of several orders of magnitude down to the Femtotesla-scale is necessary.

Given the task at hand, two of the most promising magnetometer technologies are superconducting quantum interference devices (SQUIDS) and spin-exchange relaxation free (SERF) atomic magnetometer. SQUIDs, which are used in technologies such as MRI or MEG to detect biomagnetic fields, have achieved noise levels as low as $0.3$ fT$/\sqrt{\text{Hz}}$ (Schmelz et al., 2011).

Also, Kominis et al. (2003) have presented an SERF-based atomic magnetometer with a measurement volume $1,800$ cm$^3$ and a sensitivity of $0.54$ fT$/\sqrt{\text{Hz}}$. Additionally, they have shown that the theoretical achievable fundamental sensitivity is below $0.01$ fT$/\sqrt{\text{Hz}}$. These exciting improvements however are still in the labratory phase where disturbing influences can be controlled way better than under field conditions. SQUID-based geomagnetic field sensors have been reported to have reached sensitivities of $6$ fT$/\sqrt{\text{Hz}}$ (Schönau et al., 2013). Challenges in the calibration of the sensor however limited the absolute accuracy to $0.3$ nT.

Since necessary magnetometer observations are not limited to satellite measurments, but could also originate from measuring stations deployed at ocean bottom or an array of moored bouys, there are several options to overcome the present obstacles in measurability. Given the advancements in magnetic field sensor technology, it is reasonable to assume that ENSO induced $B_r$ anomalies investigated in this study might become detectable in the intermediate future. Even more so considering that the absolute accuracy is less important than the precision in our case, since we investigate periodical variations of the geomagnetic field. Also, not only are the periodic oceanic tidally-induced radial magnetic field easily extracted from geomagnetic field observations of multiple contributing sources, but the error decreases also with $n^{-1/2}$, where $n$ equals the number of observations in a time series.

## 3.2 Spatio-temporal anomaly development

Temporal and spatial development of SST and $B_r$ anomalies of the strongest ENSO cycle of the time series (grey shaded interval in figure 1) are displayed as Hovmoeller plots in figure 2.

Considering the large scale changes in ocean temperature and salinity due to ENSO, large ocean conductance and consequently considerable $B_r$ anomalies are to be expected. In fact, the range in MaNI is approximately three times the range of the seasonal cycle in the same region (cf. 3.1). However, the changes of these surface processes are small compared to the total conductance, integrated from ocean bottom to sea surface. Hence, the anomalies in $B_r$ considered here amount to less than 1% of the total signal.

The presence of the geomagnetic equator in the equatorial Pacific region creates an additional hinderance. The vertical component of $\boldsymbol{B}_{\text{Earth}}$, the geomagnetic field component inducing $B_r$, undergoes a change of sign here and vanishes. Consequently, $B_r$ vanishes at the geomagnetic equator and expresses small values in its immediate proximity.

Also, the M2 tidal flow is not homogeneously distributed throughout the equatorial Pacific. Local maxima and minima in the tidal flow cause an additional modulation of the radial tidally-induced magnetic field.

To diminish these disturbing influences, $B_r$ anomalies have been spatially averaged from $5°$S to $5°$N. SST anomalies have also been averaged over that range for comparability of both images. Vertical lines in figure 2b at $\approx 260°$W are remaining artefacts caused by the southward dip in the geomagnetic equator in that region.

The comparison of figure 2a and figure 2b shows that positive $B_r$ anomalies emerge almost a year before they form in SST (phase I). The same is found for negative $B_r$ anomalies. They emerge months before positive SST anomalies recede and mark the end of El Niño (phase III). Although, the presented image shows only the spatio-temporal progression of the strongest cycle of an El Niño followed by a La Niña, these findings are comparable for the other regular ENSO cycles (not shown).

With the beginning of phase I, a positive $B_r$ anomaly is found west of the Niño 3.4 region, while at sea surface cold or neutral conditions are found. Then positive $B_r$ anomalies travel through the Niño 3.4 region eastwards. They are probably caused by equatorial Kelvin waves which are known to precede the onset of El Niño (Harrison and Schopf, 1984). Kelvin waves deepen the thermocline and increase the amount of warm seawater in the water column. A rise in seawater conductance, the depth-integrated seawater conductivity, is the consequence. SST anomalies have not yet formed during this phase. An intensification of positive $B_r$ anomalies on the South American west coast is observed with the arrival of the Kelvin waves several months before the ONI-defined start of El Niño. This is explainable by the deepening of the thermocline (Wang and Picaut, 2013) and the corresponding anomalous increase of warm water in the upper ocean.

During phase II, El Niño effects become apparent at the sea surface. Here, the El Niño typical weakening of trade winds leads to changes of wind patterns in the Walker Circulation and alters the equatorial ocean current system (McPhaden, 1999). As a consquence, the warm water of the Western Warm Pool flows eastward and leads to an increase and westward expansion of SST anomalies at the Peruvian coast. The eastward migrating warm water causes a thermocline shallowing in the Western Warm Pool and a simultaneous deepening of the thermocline at the Peruvian coast. This leads to negative (positive) $B_r$ anomalies west (east) of the Niño 3.4 region reaching local amplitudes of $-6\,\mathrm{pT}$ ($3\,\mathrm{pT}$) when El Niño has fully developed. The general agreement of ONI and MaNI during this phase and the co-occurring maxima of both indices show that sea surface and subsurface dynamics exhibit a similar behaviour under the influence of a common cause.

The beginning of phase III is marked by an eastward expansion of the western negative $B_r$ anomaly that has formed during phase II. The effects of El Niño, in form of warm SST anomalies, are still present for several months. Subsurface processes, however, cause an early decrease in $B_r$ anomalies and consequently in MaNI. The eastern positive $B_r$ anomaly recedes and a negative anomaly forms months before the onset of La Niña becomes apparent in ONI.

Phase IV marks the beginning of La Niña at sea surface. The Walker Circulation returns to normal conditions and the westward direction of the equatorial ocean current is re-established. Hence, the eastern thermocline shallows due to upwelling of cold water and warm surface water is transported to the western warm pool. Westward traveling SST and $B_r$ anomalies are the consequence which increases the agreement of ONI and MaNI.

With the end of phase IV a new cycle starts. The build-up of positive $B_r$ anomalies, as described in Phase I, can be observed towards the end of the plotted time interval.

The analysis shows that the identified lead in the MaNI is not just a mere forward shift of the signals but a combined result of multiple effects. We found that it is a combination of early signs for the onset of the El Niño probably caused by eastward traveling Kelvin waves and a decrease in the magnetic signal months before the actual end of El Niño due to a shallowing of the thermocline. Consequently, the cycle of $B_r$ anomalies and SST anomalies are phase-shifted.

### 3.3   Cross-correlation ONI and conductance

The findings obtained from calculating the cross-correlation between the oceanic conductance and the ONI at each grid point are summarized in figure 3.

In figure 3a, the maximum conductance anomaly of the time series at each grid point is shown. The magnitude of conductance anomalies is linked to the magnitude of relative $B_r$ changes. The largest signals are therefore evident in the Western Warm Pool and at the west coast of South America. In these regions, the thermocline undergoes the largest relative changes as a result of ENSO. We also find that the conductance anomalies are elevated in a small band throughout the whole equatorial region. This region is the passage way of the equatorial Kelvin waves which vary the thermocline depth.

In figure 3b, the maximum absolute correlation is plotted. Largest values are found east of the Niño 3.4 region. The correlation the conductance and the ONI decreases westwards in a tongue shaped pattern, like the typical SST anomalies of El Niño and La Niña.

Figure 3c shows the lag between the ONI and the conductance. The lead in conductance increases in a tongue shaped pattern originating from the South American west coast. Since the Kelvin waves travel eastwards, an increase in the lead towards their origin is a logic consequence.

For the Niño 3.4 region (solid rectangles in figure 3), we find the same characteristics as for the analysis in section 3.1. The maximum absolute correlation of the whole region ranges from $\approx 0.7$ to $\approx 0.8$ (fig. 3b). The lead distribution in the area is not uniform. A large part of the western half is leading by 5 months and decreases eastward to 2 months. The area-averaged $B_r$ anomalies of the MaNI therefore produce a signal that is leading statistically by 4 months with a correlation between 0.7 and 0.8.

## 3.4 Qualitative application of findings

The robustness of our findings is tested with a reanalysis of the correlation between ONI and MaNI using an updated averaging region for the MaNI. The new region is located at 5°N - 5°S, 150°W - 170°W. It keeps the poleward extend of $\pm 5°$ to account for an adequate averaging in consideration of the presence of the geomagnetic equator. The eastern boundary is shifted westwards to increase the lead in magnetic field anomalies over SST anomalies (see figure 3c). The westward shift is constrained by the maximum correlation found in figure 3b. A recalculation of the MaNI within the updated region is shown in figure 4.

The reanalysis shows an overall decrease in correlation between the two time series. For a lag of 0 month, the correlation is decreased from 0.63 to 0.38. The maximum correlation is decreased from 0.72 to 0.58, while the lead is increased from 4 to 5 months. Additionally, the range of the updated MaNI has reduced from $-0.84$ pT to $0.82$ pT to a range of $-0.69$ pT to $0.61$ pT. These results are in agreement with the previous findings of section 3.3. We conclude that the lead in MaNI found in section 3.1 and 3.2 is caused by the lead in conductance anomalies found in section 3.3.

Correlation and cross-correlation are a linear measure for the relation between two variables. ENSO is the defining influence on the progression of ONI and MaNI. However, we found that the processes contributing to ENSO cause differing developments in sea surface and subsurface dynamics. Consequently, the decrease in correlation should be viewed as an increase in information gained from the perspectives of SST and $B_r$ anomalies onto the same phenomenon.

## 4 Conclusions

Seawater temperature and salinity altering processes are known to be integrated in the electromagnetic oceanic tidal signals from bottom to surface. We investigated whether the tidally-induced magnetic fields could be used as an indicator for the El Niño/Southern Oscillation.

We used a coupled ocean-atmosphere general circulation model to simulate 50 years of monthly mean seawater temperature, salinity and pressure distributions. The properties were used to calculate the tidal electromagnetic signals for each month.

We analysed the relation of electromagnetic signals and ENSO by comparing two ENSO indices. These indices, calculated in the Niño 3.4 region, are the Oceanic Niño Index (ONI), based on SST anomalies, and a proposed Magnetic Niño Index (MaNI), based on anomalies in the tidal magnetic field. We show that both indices are highly correlated and MaNI is leading the ONI by 4 months.

In order to explain this lead, the spatial and temporal evolution of $B_r$ anomalies was analysed and compared with the evolution of SST anomalies. We found the lead to be explainable with eastward traveling equatorial Kelvin waves. They are known to precede the development of ENSO typical SST anomalies. They also increase the thermocline depth in the eastern Pacific ocean. In consequence, the electric conductance of the upper ocean is elevated which results in a stronger tidal magnetic field.

Based on these results, we analysed the relation between the ONI and equatorial Pacific conductance anomalies. The spatial distributions of correlation, lead and signal strength were in good agreement with the found MaNI characteristics. We showed that correlation of conductance anomalies and ONI increases eastward, while the lead over the ONI is increased westward.

With these findings we updated the averaging-region for the recalculation of the MaNI. With the new index, our interpretation was confirmed and the lead in MaNI was increased to 5 months. At the same time, signal strength and correlation were reduced. The decrease in correlation is interpreted as a gain in information about subsurface dynamics of ENSO rather than a loss in information about ENSO itself. Traditionally, researchers have focused on sea surface dynamics for signs of El Niño. The latest research, however, shows that subsurface dynamics play a crucial role in the build-up phase and the decline of El Niño. An increased focus on subsurface processes is therefore necessary to understand the El Niño/Southern Oscillation completely.

With the modeled tidal magnetic field anomalies being too small to be detectable with contemporary measuring methods, the presented results are not applicable at present times. However, magnetometer sensitivity under labratory conditions have reached noise levels several orders of magnitude below the necessary detection threshold for the presented $B_r$ anomalies. Consequently, a detection of the analysed signals is at least theoretical possible, even though it might be improbable due to technical limitations in field measurements.

In summary, our study shows that the dynamic of tidally-induced radial magnetic field anomalies contains information for an early awareness of developing anomalous warm and cold ENSO conditions. In the case of substantial improvements in observation techniques so that measuring these signals may be realised, this may be used to improve current warning systems. Consequently, socio-economic effects brought into different regions of the Earth due to ENSO's teleconnections could be diminished with early pro-active counter measurements.

*Author contributions.* J.S. developed the concept for the study. T.W. performed the climate-experiment providing the necessary ENSO data. J.P., J.S. and C.I. designed all numerical experiments which where performed by J.P. The manuscript was written by J.P. with the assistance of J.S., C.I., T.W. and M.T. All authors discussed the results and commented on the manuscript.

*Competing interests.* The authors declare no competing financial interests.

5  *Acknowledgements.* This study is funded by the German Research Foundation's priority programme 1799 'Dynamic Earth' and by the Helmholtz Association of German Research Centres. We also want to thank the Max Planck Institute for Meteorology for making their climate model ECHAM6/MPIOM available to the research community. The model experiments were carried out an the supercomputing system of the German Climate Computation Centre (DKRZ) in Hamburg, Germany. We acknowledge the generation and distribution of the TPXO tidal data. Last but not least, we want to thank Alexey Kuvshinov for kindly providing his 3D EM induction code and the model of
10  the mantle conductivity. The work would not have been able without his training and help. Thank you.

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

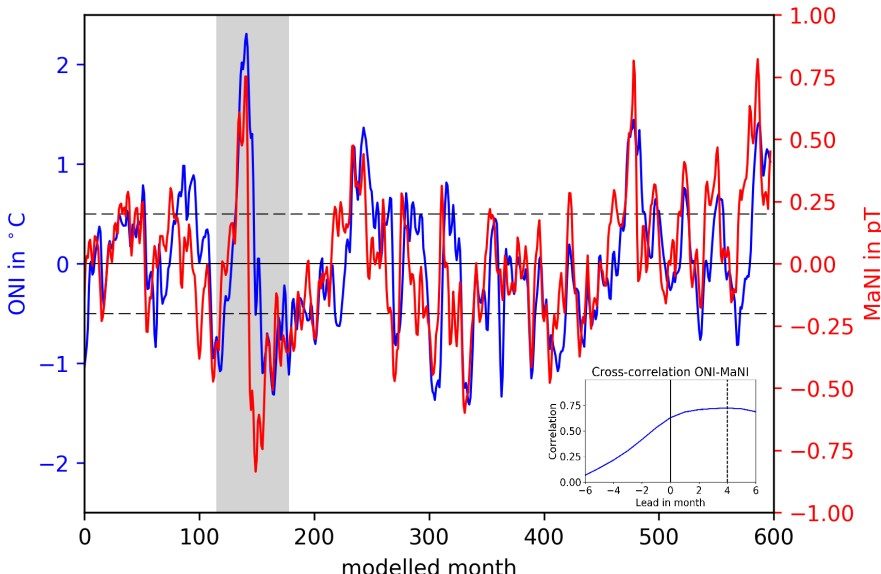

**Figure 1.** ENSO indices. ONI derived from sea surface temperatures (blue curve) and MaNI derived from the radial tidally-induced magnetic field $B_r$ (red curve). The dashed lines mark the threshold of $\pm 0.5\ ^\circ$C, the threshold for El Niño and La Niña events. The grey shaded area marks the strongest cycle of ENSO events (used for further analysis). The embedded plot shows the cross-correlation between ONI and MaNI. For positive leads, MaNI leads ONI.

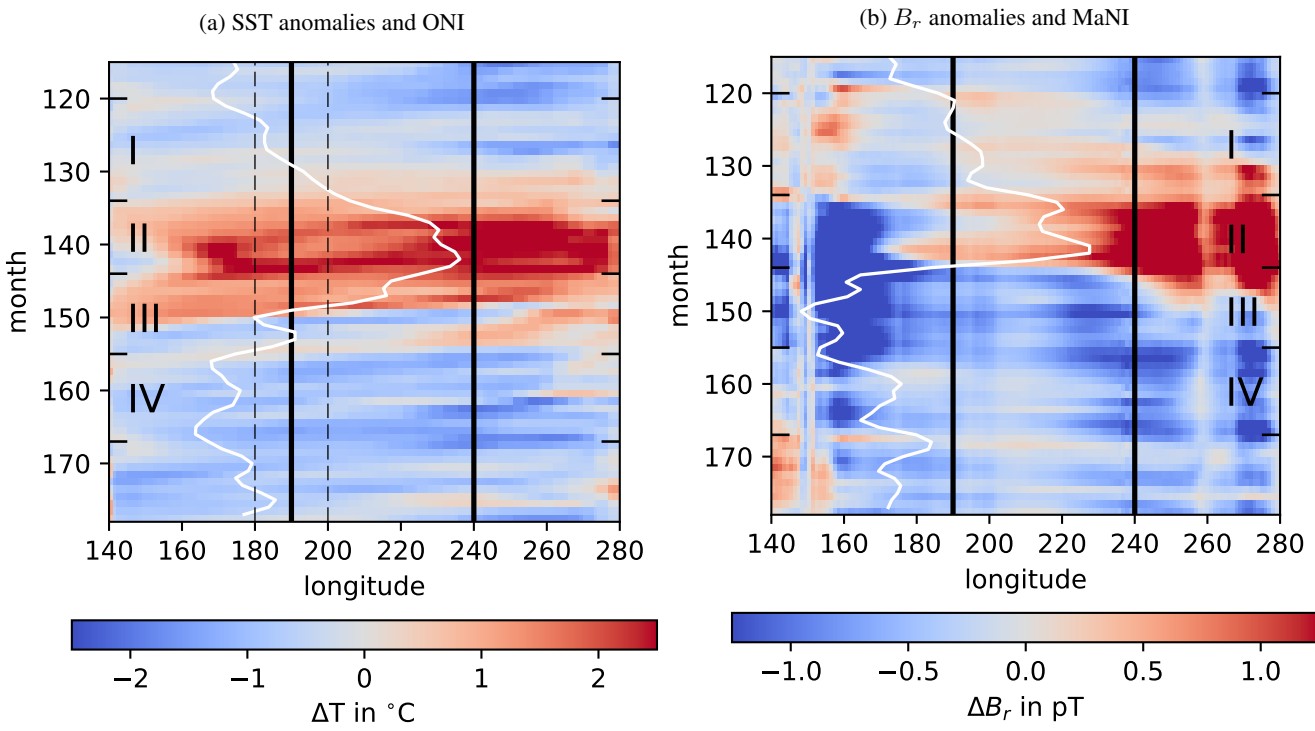

**Figure 2.** Hovmoeller plots of sea surface temperature anomalies (left) and $B_r$ anomalies (right) averaged from $5°$S to $5°$N. The time interval shows the strongest ENSO cycle cf. figure 1. Vertical black lines enclose the Niño 3.4 region used to calculate ONI and MaNI. The solid white lines represent the indices derived from the individual anomalies centred on $170°$E ($20°$ of longitude correspond to $1°$ C (left) and $0.4$ pT (right)). The dashed lines represent the ONI thresholds for El Niño/ La Niña of $±5$ $°$C (left).

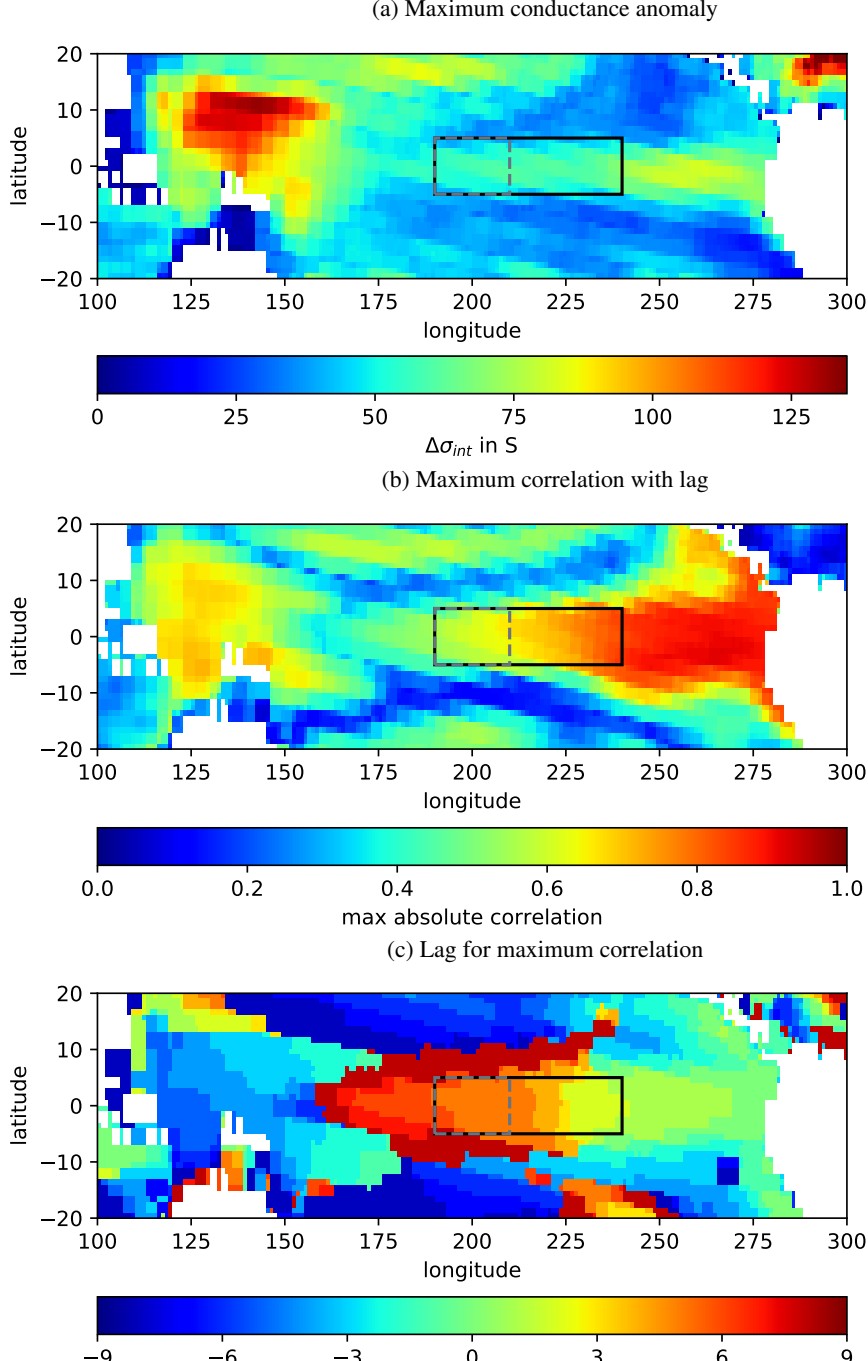

**Figure 3.** Cross-correlation analysis between the ONI and the conductance ($\sigma_{int}$) at each grid point: maximum absolute conductance anomaly (top), absolute maximum correlation, the peak value of the cross-correlation (middle), corresponding lead/lag to the absolute maximum correlation (bottom). The solid rectangle shows the location of the Niño 3.4 region, the dashed rectangle shows the location of an updated MaNI (5°N - 5°S, 150°W - 170°W).

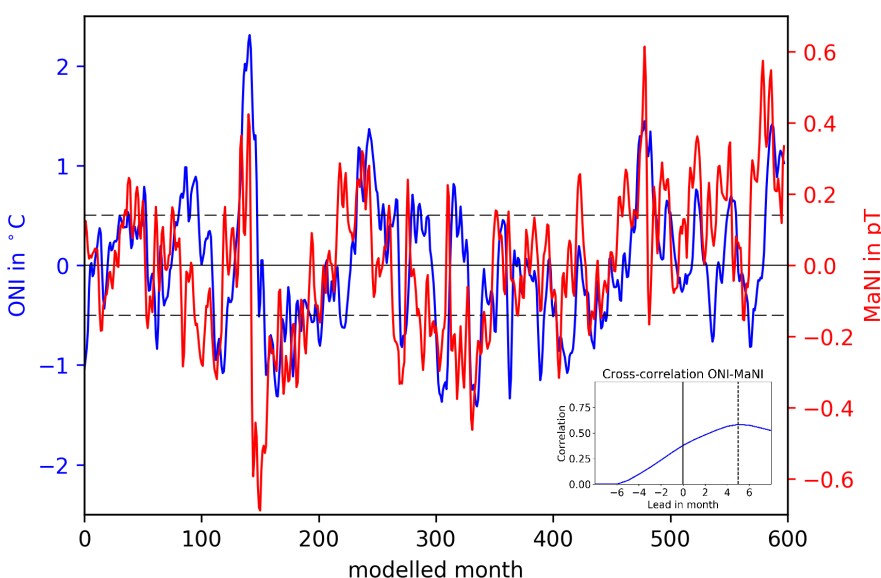

**Figure 4.** Comparison of time series of ONI (blue) and updated MaNI (red). Anomaly strength and correlation are reduced, while the lead is increased.