# Peer review of "Electromagnetic characteristics of ENSO"

_Ocean Science, 2018_

## Referee Comment (RC1) · Anonymous Referee #1 · 6 Apr 2018

Summary of results

The paper describes an application of established electromagnetic theory to use variations in the tidally-induced magnetic field to potentially identify anomalies in the conductivity in the Equatorial Pacific Ocean, and use these as indicators of incipient El Nino/La Nina (ENSO) events. They show acceptable levels of correlation between these signals and ENSO events with a lead time of several months, and state that this provides a higher level of predictability than observations of surface fields.

General style

The paper is generally well structured and clearly written, and the standard of English is good.

[Figure]

General comments

There are in my opinion some logically important steps missing in sections 2.1 and 2.1, which purport to explain how the interaction between tidal motions, the Earth's magnetic field and the local T/S structure in the tropical Pacific gives rise to measurable magnetic field anomalies.

The results described in the paper from the ocean model are very interesting, but I have some difficulty with the overall message of this work. There is a bald statement in the Conclusions section that the estimated magnetic anomalies are currently too small to measure, which would seem to invalidate the main conclusion (stated directly in the final sentence of the paper) that this technique might be used to predict the onset of an ENSO event. This, in my opinion, demands some discussion of how useful the technique actually is, both at the present and at some time in the future. Firstly, is there a fundamental limit to the smallest measurable magnetic signal in the ocean? How large is the effect of internal waves and other small-scale oceanic motions on the magnetic anomaly, compared with that due to the predictable tidal signal? Are there any foreseeable improvements in technology that would reduce the SNR to enable these signals to be detected? Secondly, as the authors note in the Introduction, upper ocean heat content is already known to be a good predictor of ENSO: surely this is a far easier precursor to measure directly than the admittedly tiny magnetic field anomalies? Are there any real advantages of the technique discussed in this paper over conventional monitoring of standard hydrographic fields (e.g. the TAO/TRITON array)?

Recommendation

Accept with major revisions

Specific comments

Abstract

"Tidally-forced" is more correct than "tidal-forced", which occurs throughout the paper,

particularly in the Abstract.

P1L2: Replace "separable from" with "distinguishable in"

1. Introduction

P2L9: "thermohaline circulation" does not drive the ocean – replace this with "buoyancy fluxes"?

P2L31: "pacific" should have a capital.

P2L5: What does "variating" mean?

There is usually a short description of the structure of the rest of the paper at the end of the Introduction, but one is lacking here. In the present case I think this would be very helpful, since it would steer the reader through the logical chain of argument from tidal currents to the at least theoretically detectable magnetic anomalies, and hence to correlations between the latter and ENSO events.

2.1 Ocean and tidal induced currents

P3L26: Insert "with" after "linearly vary".

P3L27: What does "well know for real observation times" mean?

2.2 EMOTS

Perhaps I am not paying enough attention here, but there doesn't seem to be a clear link between the induced currents jm2 in the previous section and the measurable resulting magnetic field. Specifically, Br is referred in the subsequent section as being defined in the present section, but it is not. There is quite a dense description of the conductivities, but this doesn't seem to go anywhere – is there some text missing here?

Also, are the EMOTS measurable in the magnetic or electric field? Presumably the magnetic component, since this is referred to several times further on, but this is not stated explicitly here.

Finally, I think this section would be a suitable place for an error analysis of the pre-dicted signals. Specifically, how large is the contribution of the other, more stochas-tic, flow components (e.g. tropical instability waves, equatorial Kelvin waves, internal waves, etc) relative to the tidal signal? What is the magnitude of variability of the local geomagnetic field? What are the current measurement limits?

3.2 Spatio-temporal anomaly development

P6L5: To make this paragraph make more sense, perhaps state explicitly that Bearth vanishes at the geomagnetic equator. Also, say explicitly how the spatial structure of the tidal flow reduces Br in this region. Presumably this vanishing of the background field implies that equatorial regions are the worst place to use this technique!

P6L8: Figure 2b strictly only shows one anomaly, rather than a robust lead-lag rela-tionship between "anomalies", as suggested in the text

It would be helpful to discuss here how the computed Br anomalies relate to the noise levels and to instrumental measurement capabilities.

4 Conclusions

P8L18: The statement that the predicted Br anomalies are too small to be detected with current observational techniques is quite startling, but is not discussed further. In fact, the following text then states how potentially useful the results of the paper are in predicting ENSO, despite them being at present totally impracticable! It would be useful here to include something about at least about the possibility of improving the detection threshold – is this at least theoretically possible?

---

## Referee Comment (RC2) · Anonymous Referee #2 · 13 Apr 2018

The authors present results of 3-D EM modelling aiming to estimate variability of semi-diurnal lunar (M2) tidal radial magnetic field signals due to ENSO. By the way, authors did not specify in the paper an observation level at which these results are obtained but I presume that it is a satellite (430? km) altitude. As expected, the signature appeared to be super tiny (fractions of picoteslas), and I am very doubtful that one will be able to detect these signal from satellite data even in the far future. The main reason is that satellites measure a mixture of signals from different sources and there is no any chance to separate the signals with such level of accuracy. However I think the paper could be published if the authors will moderate their claims, for example, stating that it is just "in silico" study which most probably is not of practical use. My other comments are summarized below.

[Figure]

1. Page 1: 3D induction model -> 3D EM induction solver;

2. Page 2: Sentence about finding of Manoj et al (2006) about ENSO and ACC. Please elaborate, it reads weird;

3. Page 3: theta is co-latitude?

4. Page 4: the induction model -> 3D EM induction solver;

5. Page 4: Please update the next sentence. "The solver is based on a contracting volume integral equation approach (Pankratov et al., 1995; Singer and Fainberg, 1995).

---

## Author Comment (AC1) · 23 May 2018

We thank anonymous Referee 2 for his/her review and for providing helpful remarks on our manuscript which helped to improve and clarify our work greatly.

- "*By the way, authors did not specify in the paper an observation level at which these results are obtained ...* "

- "*I am very doubtful that one will be able to detect these signal from satellite data even in the far future. I think the paper...*"

- " *I think the paper could be published if the authors will moderate their claims, for example, stating that it is just "in silico" study which most probably is not of practical use.*"

[Figure]

We added the observation level, which actually is at sea level, where $B_r$ exhibits a finer lateral structure. Also, we did not want to constrain the findings to satellite observations. Either way, it will be a challenging task to observe the presented signals. But, we believe that the unique selling point of the electromagnetic oceanic tidally-induced signals are their well known frequencies. We think that through signal processing even weak periodical signals can be extracted from noisy observations, provided that the time series of observations is sufficiently long. We agree, however, that detecting these signals with satellites, the currently used technique to detect the radial magnetic field component of oceanic tidally-induced magnetic fields, in the intermediate future will be very challenging and have moderated our claims accordingly.

All technical notes and suggestions have been fully implemented.

---

## Author Comment (AC2) · 23 May 2018

We thank the reviewer for carefully reading our manuscript and providing many helpful scientific and technical remarks. The resulting changes improved the manuscript greatly.

**General comments:**

-"*There are in my opinion some logically important steps missing in sections 2.1 and 2.1, which purport to explain how the interaction between tidal motions, the Earth's magnetic field and the local T/S structure in the tropical Pacific gives rise to measurable magnetic field anomalies.*"

We improved and restructured the mentioned sections in order to add more background

on EMOTS and to clarify the link between the induced electrical current and the resulting electromagnetic signals. Also, we tried to focus on the advantage of the strict periodicity and the consequent detectability which distinguishes the oceanic tidally-induced signals from signals induced through other oceanic processes.

-"*Firstly, is there a fundamental limit to the smallest measurable magnetic signal in the ocean? How large is the effect of internal waves and other small-scale oceanic motions on the magnetic anomaly, compared with that due to the predictable tidal signal? Are there any foresee- able improvements in technology that would reduce the SNR to enable these signals to be detected?*"

In order to address this important issue, we discussed the theoretical measurability and recent advances in the magnetometer technology in section 3.1. We think that the added context will help to clearly classify the presented model study as currently impracticable but with possible future applications.

-"*Secondly, ... Are there any real advantages of the technique discussed in this paper over conventional monitoring of standard hydrographic fields (e.g. the TAO/TRITON array)?*"

Given the current situation of not being measurable, we refrain from an in-depth discussion of this issue in the paper. However, we addressed this issue briefly in the introduction.

The oceanic tidally-induced magnetic fields, detectable in Swarm observations, are able to provide information of the ocean from sea surface to ocean bottom and have therefore several applications appart from detecting ENSO. We believe that the presented study, if applicable in the future, would be a complementary technology to the existing techniques.

**Specific Comments:**

We included all specific comments proposed by the rewiever. In the following, we
respond specifically to the issues related to substantial manuscript adoptions in section 2.2, 3.1, 3.2 and 4.

2.2 EMOTS

-"*Perhaps,... there doesn't seem to be a clear link between the induced currents $j_{M2}$ in the previous section and the measurable resulting magnetic field...*

-"*Also, are the EMOTS measurable in the magnetic or electric field? ... this is not stated explicitly here.*"

We implemented necessary changes to answer these questions. Also, we addressed this issue in our answer to the general comments (please see above).

-"*Finally, I think this would be a suitable place for an error analysis of the predicted signals. ...how large is the contibution of the other, more stochastic flow components ... relative to the tidal signal?*"

This is a very interesting question. More stochastic components have the disadvantage of not having a distinct frequency which would allow a straightforward extraction from magnetometer observations. Therefore, their induced magnetic field may not be detectable directly. Their indirect effect on the T/S structure and the resulting conductivity on the other side may be detectable through variations of EMOTS. Their contribution can be expected to be in the same order of relative conductance changes in the water column. Given the ratio between ocean depth and total change in conductivity, the resulting effect onto the conductance (integrated conductivity) and therefore the EMOTS may be minute.

We addressed this issue briefly in the updated manuscript.

-"*What magnitude of variability of local geomagnetic field?*"

The influence of the secular variation onto the EMOTS has not been investigated so far. Previous studies have neglected this effect even on longer time-scales under the

assumption that $B_{Earth}$ is very well known and the linear contribution of $B_{Earth}$ to $j_{M2}$ can be removed accordingly. However, given the necessary precision needed to detect the presented signals, a subsequent study investigating this issue might be necessary.

-"*What are the current measurement limits?*"

Please see below.

3.2 Spatio-temporal anomaly development

-"*It would be helpful to discuss here how the computed $B_r$ anomalies relate to the noise levels and to instrumental measurement capabilities*"

We realize that the issue may have been addressed too briefly in section **3.1**. That's why we rearranged this section and discussed this issue in more detail. We included recent advances compared to current measurement limits of the Swarm satellites. Furthermore, we discussed that, despite those recent advances, an actual implementation of the presented techniques requires further advances for an in-field applicable technical realization. We chose this section, since we mention the magnitude of the signals here for the first time.

4 Conclusion

-"*...It would be useful here to include something about at least about the possibility of improving the detection threshold – is this at least theoretically possible?*"

In addition the changes in section 3.1 (please see above), we changed this section accordingly and moderated the previous claims.